# Yogurt Produced by Novel Natural Starter Cultures Improves Gut Epithelial Barrier In Vitro

**DOI:** 10.3390/microorganisms8101586

**Published:** 2020-10-15

**Authors:** Nikola Popović, Emilija Brdarić, Jelena Đokić, Miroslav Dinić, Katarina Veljović, Nataša Golić, Amarela Terzić-Vidojević

**Affiliations:** Institute of Molecular Genetics and Genetic Engineering, University of Belgrade, Vojvode Stepe 444a, 11042 Belgrade, Serbia; popovicnikola@imgge.bg.ac.rs (N.P.); emilija@imgge.bg.ac.rs (E.B.); jelena.djokic@imgge.bg.ac.rs (J.Đ.); mdinic@imgge.bg.ac.rs (M.D.); katarinav@imgge.bg.ac.rs (K.V.); natasag@imgge.bg.ac.rs (N.G.)

**Keywords:** *Streptococcus thermophilus*, *Lactobacillus bulgaricus*, yogurt, tight junctions, autophagy

## Abstract

Yogurt is a traditional fermented dairy product, prepared with starter cultures containing *Streptococcus thermophilus* and *Lactobacillus bulgaricus* that has gained widespread consumer acceptance as a healthy food. It is widely accepted that yogurt cultures have been recognized as probiotics, due to their beneficial effects on human health. In this study, we have characterized technological and health-promoting properties of autochthonous strains *S. thermophilus* BGKMJ1-36 and *L. bulgaricus* BGVLJ1-21 isolated from artisanal sour milk and yogurt, respectively, in order to be used as functional yogurt starter cultures. Both BGKMJ1-36 and BGVLJ1-21 strains have the ability to form curd after five hours at 42 °C, hydrolyze α_s1_**-**, β**-**, and κ**-** casein, and to show antimicrobial activity toward *Listeria monocytogenes*. The strain BGKMJ1-36 produces exopolysaccharides important for rheological properties of the yogurt. The colonies of BGKMJ1-36 and BGVLJ1-21 strains that successfully survived transit of the yogurt through simulated gastrointestinal tract conditions have been tested for adhesion to intestinal epithelial Caco-2 cells. The results reveal that both strains adhere to Caco-2 cells and significantly upregulate the expression of autophagy-, tight junction proteins-, and anti-microbial peptides-related genes. Hence, both strains may be interesting for use as a novel functional starter culture for production of added-value yogurt with health-promoting properties.

## 1. Introduction

Yogurt is popular dairy product obtained by fermentation of lactose to lactic acid by lactic acid bacteria (LAB), and can be made from all types of milk [1,2]. As specified by Codex Alimentarius Standard No. 243/2003, “yogurt culture” is composed of *Streptococcus thermophilus* and *Lactobacillus delbrueckii* subsp. *bulgaricus* that are responsible for formation of typical yogurt flavor during sugar fermentation (glycolysis), proteins degradation (proteolysis), and fat degradation (lipolysis) [3]. They also produce exopolysaccharides (EPSs) that provide viscous texture of yogurt, and have a beneficial effect on the consumer’s health [4]. In addition, as a dairy product, yogurt is an excellent source of vitamins, minerals, and calcium necessary for healthy teeth, bones, and immune system [5]. Besides the presence of *S*. *thermophilus* and *L. bulgaricus* in starter cultures, yogurt-like products may contain other LAB strains with functional probiotic properties [6]. The prerequisite for use of LAB strains in dairy starter cultures is their safety qualified presumption of safety (QPS) status recommended by the EFSA (2018) [7]. Moreover, starter cultures should be stable, and able to survive at refrigerator temperature during the storage period (usually 21–28 days). It was noticed that people who suffer from lactose intolerance can tolerate yogurt much better than milk, since yogurt cultures improve lactose digestion and eliminate symptoms of intolerance [8]. Hence, yogurt culture obtained the health claim from The Panel on Dietetic Products, EFSA for “lactose digestion”, providing the health benefits to the host [9]. According to FAO–WHO (2002), “live microorganisms, which when administered in adequate amounts confer a health benefit on the host” are considered as probiotics [10]. FAO–WHO also defined that probiotics should be non-pathogenic, susceptible to clinically relevant antibiotics, should survive simulated gastrointestinal tract (GIT) conditions, and have the ability to colonize GIT epithelium of the host and to inhibit the growth of pathogenic and spoilage microorganisms [9,11,12,13,14]. In recent years, whole-genome sequence data has been increasingly used to identify potential probiotic strains, as well as to characterize strains in terms of their potential functional characteristics of importance [7,15]. Importantly, probiotics have a role in gut microbiota stabilization after various forms of diarrhea, or after prolonged use of antibiotics [16,17]. In line with that, daily consumption of yogurt containing an adequate amounts of live LAB can exert positive effects on gut microbiota [18]. Moreover, yogurts produced with probiotic LAB have shown an inhibitory effect on colon cancer, the ability to restore the gut homeostasis, preventing the development of various inflammatory bowel diseases, such as ulcerative colitis and Chron’s disease, and mitigating the adverse effects of these diseases [19,20]. Indeed, the intestinal epithelium is shown to be the primary target for beneficial LAB. Recent data showed that LAB-associated biomolecules and metabolites could trigger autophagy [21,22,23], an important mechanism of cell recycling, required for anti-microbial protection, gut ecology regulation, appropriate intestinal immune response, and maintenance of epithelial barrier integrity [24]. Having in mind that all of these processes are regulated by autophagy, its modulation by LAB could represent a new strategy in maintenance of intestinal mucosal physiology.

Lactic acid bacteria are widespread in nature, but they are predominant in dairy products, especially those manufactured from raw milk. Indigenous microbiota of raw milk have a great impact on the specific sensory properties of raw milk products [25]. There is a recommendation to use starter culture in the production under aseptic condition in order to obtain high and constant quality of products for health consumers [9]. Many new LAB strains, so-called “wild strains”, isolated from raw milk and raw milk products offer new opportunities for research and application in practice.

Considering the limited biodiversity of commercial starter cultures, there has been a trend of selection and development of new starter cultures from the autochthonous microbiota, isolated from artisanal spontaneous fermented food products and their use in industrial production of fermented foods. The use of such newly-isolated and well-characterized cultures aimed to produce fermented foods in a more controlled way, with similar sensory characteristics and nutritional value to the traditional products, even with special health-promoting properties [26,27,28,29].

In line with this, the aim of the present study was technological and probiotic characterization of *S*. *thermophilus* BGKMJ1-36 and *L. bulgaricus* BGVLJ1-21 strains isolated from raw milk artisanal sour milk and yogurt, respectively. We show that yogurt produced with these strains as a starter, at laboratory scale, possesses excellent probiotic properties, with the potential to modulate gut autophagy and strengthen the intestinal epithelial barrier.

## 2. Materials and Methods

### 2.1. Origin of Strains Streptococcus thermophilus BGKMJ1-36 and Lactobacillus bulgaricus BGVLJ1-21 and Growth Conditions

For this study, two thermophilic LAB strains from the bacterial collection of the Laboratory for Molecular Microbiology, Institute of Molecular Genetics and Genetic Engineering, University of Belgrade, Serbia were used. *Streptococcus thermophilus* BGKMJ1-36 was isolated from artisanal sour milk (traditionally manufactured in household in the village Jabuka, near the city Prijepolje, Serbia, altitude 1300 m), while *L. bulgaricus* BGVLJ1-21 was isolated from artisanal yogurt (traditionally manufactured in household in the village Mlečiške Mehane, near the Vlasina Lake, Serbia, altitude 1307 m) [29]. Both strains are deposited in the Belgian Coordinated Collections of Micro-organisms/Laboratorium voor Microbiologie ‒ Bacteriënverzameling (BCCM/LMG), University of Ghent, Belgium under accession numbers LMG P-31742 for BGKMJ1-36 strain, and LMG P-28578 for BGVLJ1-21 strain.

The optimal growth of the strain BGKMJ1-36 is in M17 medium of pH 7.2 (Merck GmbH, Darmstadt, Germany), supplemented with 0.5% (*w*/*v*) glucose (GM17) at 37 °C under anaerobic conditions, while BGVLJ1-21 grows anaerobically in MRS medium of pH 5.7 (Merck). Anaerobic conditions were obtained in a CO_2_ incubator (HERAcell 150, Thermo Electron LED GmbH, Langenselbold, Germany) with 5% CO_2_. BGKMJ1-36 and BGVLJ1**-**21 strains were stored at –80 °C in GM17 and MRS broths supplemented with 15% (*w*/*v*) glycerol, respectively.

### 2.2. Physiological, Biochemical, and Technological Characterization of Streptococcus thermophilus BGKMJ1-36 and Lactobacillus bulgaricus BGVLJ1-21 Strains

The strains BGKMJ1-36 and BGVLJ1-21 were subjected to a set of biochemical and physiological tests as follows: growth at different temperatures (15 °C, 37 °C, and 45 °C); growth in broth with 2% NaCl (*w*/*v*); L**-**arginine hydrolysis, citrate utilization as energy source [30]; CO_2_ production from glucose in reconstituted MRS broth tubes containing inverted Durham bells; the production of acetoin by the Voges–Proskauer test [31]; and diacetyl production (qualitatively). After overnight incubation of the strains inoculated in 11% reconstituted skimmed milk (RSM) at 37 °C, 1 mL of coagulated milk was mixed with 0.1 g of creatinine (Alfa Aesar, GmbH & Co KG, Karlsuhe, Germany) and 1 mL of 30% NaOH (*w*/*v*). Diacetyl production was scored as formation of a red halo at the top of the tubes after 2 h of incubation at room temperature. EPS production was detected visually (on GM17 or MRS agar plates, depending on the strain) as long strands when the colonies were extended with an inoculation loop [32]. Speed curdling was determined visually by time, measuring from the moment of inoculation of 11% sterile RSM with 3% of each single starter culture grown up in RSM to the moment of curd forming at the incubation temperature of 42 °C. Aggregation of tested starter cultures was detected visually after shaking of tubes with inoculated GM17 or MRS broth that were previously incubated overnight at 37 °C.

### 2.3. Safety Assessment

#### 2.3.1. Hemolytic and Gelatinase Activity Assays

Hemolytic activity was determined on Columbia Blood Agar containing 5% (*v*/*v*) defibrinated horse blood (Torlak, Belgrade, Serbia). After 48 h of incubation at 37 °C, hemolytic activities were detected as halo zones around the colony [33]. Gelatinase activity was determined on agar plates containing 3% (*w*/*v*) gelatine (Difco, Becton Dickinson, Sparks, Meryland, USA). After 48 h of bacterial cultivation, plates were filled with 550 g/L ammonium sulphate [34], and gelatinase activity was detected as halo zone. The absence of such zone was considered as absence of gelatinase activity. For both assays, the positive control was *Enterococcus faecalis* V583.

#### 2.3.2. Antibiotic Susceptibility Testing

Minimal inhibitory concentrations (MICs) were determined by microdilution testing, following European Food Safety Authority criteria [7]. Susceptibility was tested against: ampicillin (2 mg/L), vancomycin (4 mg/L), gentamicin (32 mg/L), streptomycin (64 mg/L), erythromycin (2 mg/L), clindamycin (2 mg/L), tetracycline (4 mg/L), and chloramphenicol (4 mg/L) for *S. thermophilus*, while for *L. bulgaricus,* it was tested against ampicillin (2 mg/L), vancomycin (2 mg/L), gentamicin (16 mg/L), kanamycin (16 mg/L), streptomycin (16 mg/L), erythromycin (1 mg/L), clindamycin (4 mg/L), tetracycline (4 mg/L), and chloramphenicol (4 mg/L). Microdilution tests were performed in Hi-Sensitivity Test Broth (HiMedia, Mumbai, India). The final CFU per well was 5 × 10^6^. Cell density was monitored after 24 h incubation at 37 °C at 595 nm using spectrophotometer Plate Reader Infinite 200 pro (MTX Lab Systems, Vienna, Austria). *En. faecalis* V583 [35], *S. thermophilus* BGVLJ1-44, and *Lactobacillus helveticus* BGRA43 [36] were included as a quality control strains (control of antibiotic potency and quality of medium).

### 2.4. Antimicrobial Activity 

The antimicrobial activity of *S*. *thermophilus* BGKMJ1-36 and *L. bulgaricus* BGVLJ1-21 was tested on various indicator strains: *Lactobacillus plantarum* A112, *Lactobacillus casei* BGHN14, *Lactococcus lactis* subsp. *lactis* BGMN1**-**596, *Lactococcus lactis* subsp. *lactis* BGZLM1**-**24, *Lactococcus lactis* subsp. *cremoris* NS1, *Enterococcus faecalis* BG221, *Listeria monocytogenes* ATCC 19111, *Escherichia coli* ATCC 25922, and *Salmonella* Enteritidis 654/7E by modified agar well diffusion assay [37]. Briefly, after incubation at 37 °C for 16 h, soft GM17 and MRS agars (0.7% *w*/*v*) containing lactococci, enterococci, or lactobacilli indicator strains were overlaid onto respective GM17 and MRS plates. The plates were incubated overnight at 37 °C. A clear zone of inhibition of indicator strain growth around the well was taken as a positive signal for antimicrobial activity. A crystal of protease TYPE XIV (Sigma Chemie GmbH, Deisenhofen, Germany) was placed close to the edge of the well containing the overnight culture, to confirm the production of bacteriocin-like antimicrobial compounds of proteinaceous nature.

### 2.5. Proteolytic Activity

Both strains, BGKMJ1-36 and BGVLJ1**-**21, were assayed for proteolytic activity, as previously described [38]. The collected fresh cells (10 mg with an approximate density of 10^10^ cells/mL) were resuspended in 0.1 M of sodium phosphate buffer (1 M NaH_2_PO_4_ and 1 M Na_2_HPO_4_) with pH of 6.8, and mixed in a 1:1 ratio with 5 mg/mL of α_s1_**-**, β**-**, and κ**-** casein, respectively (Sigma, St. Louis, MO, USA), dissolved in the identical buffer. The mixtures were incubated for 4 h at 42 °C. The degradation of α_s1_**-**, β**-**, and κ**-** casein was analyzed on 12.5% sodium dodecyl sulphate-polyacrylamide gel electrophoresis (SDS-PAGE).

### 2.6. Yogurt Manufacturing 

Preparation of yogurt included milk fermentation process by use of defined starter cultures in the optimal ratios according to Dirar (1993) [39], with some modifications. Briefly, autoclaved RSM milk was inoculated with 2% of each overnight probiotic cultures grown in GM17 broth (BGKMJ1-36) and MRS broth (BGVLJ1**-**21), respectively. These single RSM milks inoculated with BGKMJ1-36 and BGVLJ1**-**21 strains were incubated overnight at 37 °C. Yogurt was prepared in pasteurized milk (Imlek, Belgrade, Serbia). Separately, BGKMJ1-36 and BGVLJ1-21 starter cultures were added in pasteurized milk, in an amount of 3% of the total milk amount in the ratio 1:2. The total amount of yogurt starter culture for milk inoculation was 3% of total milk amount. The starter cultures were added in milk, separately, in optimal ratios: BGKMJ1-36:BGVLJ1**-**21 = 1:2. Incubation of inoculated milk was carried at 42 °C for 4 to 5 h until the pH value was lowered to about 4.8. After that, the glass bottles with fermented milk were rapidly cooled to 15 °C, and then shaken to obtain a homogenized and consistent structure of yogurt. Then, the cooling of the yogurt was continued in the refrigerator to 4 °C, where the pH value of yogurt was further slowly lowered to 4.6 during cooling. The pH value and total viable count of yogurt bacteria was estimated immediately after inoculation of milk, as well as each hour during the first 4 to 5 h of the milk fermentation, and after 1, 7, 14, 21, and 28 days of storage of fermented milk at 4 °C. The yogurt was previously aliquoted in separate batches, so that oxygen uptake was not possible during the storage. For each measurement of the pH value and the total number of viable bacteria, during the whole storage period, a separate batch was provided. Experiments were performed in three independent measurements.

### 2.7. Survival in Simulated Gastrointestinal Tract Conditions

Survival in chemically simulated GIT conditions was performed using an in vitro test, as described previously [40]. Viable cell counts were recovered from the one-day-old yogurt, after 90 min of gastric juice (125 mM NaCl, 7 mM KCl, 45 mM NaHCO_3_, 0.3% pepsin (Sigma), pH 2) challenge, 10 min of duodenal juice (1% bile salt (Sigma), pH 8) challenge, and 120 min of intestinal juice (0.3% bile salt, 0.1% pancreatin (“Pancreas acetone powder porcine Type I”, Sigma), pH 8) challenge, respectively. We used GM17 for BGKMJ1-36 and MRS for BGVLJ1-21 colony counts. Results were expressed as CFU/mL of survived cells, and calculated from the viable counts recovered after every challenge of simulated GIT condition, with respect to the initial counts. Experiments were carried out in triplicate.

### 2.8. Cell Culture and Treatments

Differentiated human enterocyte-like Caco-2 cells were used as an in vitro small intestine epithelial barrier model. The cells were grown in Dulbecco’s Modified Eagle Medium (DMEM) supplemented with 10% fetal bovine serum (FBS), 100 μg/mL streptomycin, 100 U/mL penicillin, and 2 mM L-glutamine (Gibco). The cells were maintained in 75 cm^2^ flasks at 37 °C in a humidified atmosphere containing 5% CO_2_. Caco-2 cells were seeded in a 24-well plate (2 × 10^5^ cells/well), and incubated at 37 °C for 21 days to allow differentiation (reaching the number of 2 × 10^6^ of Caco-2 cells/well). After the indicated time, the cells were treated for 2 h with 50 µL of the product (containing 4.66 × 10^5^ of live BGKMJ1-36, 4.5 × 10^5^ of live BGVLJ1-21, and damaged bacterial particles) obtained by GIT treatment of 10^8^ live bacteria in yogurt (multiplicity of infection of 100). Before the treatment, GIT product was washed and resuspended in phosphate-buffered saline (PBS). After the treatment, cells were washed three times with PBS to remove the non-adherent bacteria, followed by a trypsinization step in order to detach cells (Trypsin-EDTA, Torlak) for the assessment of BGKMJ1-36 and BGVLJ1-21 adhesion properties [41]. Simultaneously, the cells were collected and stored at -80 °C for total RNA isolation and qPCR analysis.

### 2.9. Cytotoxicity Assay

A lactate dehydrogenase (LDH) Cytotoxicity Assay Kit (Thermo Fisher Scientific, Waltham, Massachusetts, USA) was used to evaluate the level of cytotoxicity of yogurt applied on Caco-2 cells. After two hours of treatments, supernatants were collected, and released LDH was detected by following the manufacturer’s instructions. The absorbance was measured at 450 nm using the Plate Reader Infinite 200 pro (MTX Lab Systems).

### 2.10. RNA Isolation and qPCR

Total RNA was extracted from Caco-2 cells according to the protocol described in [42]. All samples were treated with DNase I, using the Ambion DNA-free™ Kit (Thermo Fisher Scientific) to remove DNA contamination from RNA samples. Reversed transcription was done with the RevertAid RT kit, using 1 µg of isolated RNA as a template, according to the manufacturer′s protocol (Thermo Fisher Scientific). Synthesized cDNA was amplified in a 7500 real-time PCR system (Applied Biosystems), using SYBR™ Green PCR Master Mix (Applied Biosystems) under the following conditions: 10 min at 95 °C activation, 40 cycles of 15 s at 95 °C, and 60 s at 60°C. Normalization was done against the *GAPDH* gene, using the 2^-ΔΔCt^ method [43]. Primers were designed based on sequences available in the NCBI database by utilizing the NCBI Primer-Blast tool, available online (https://ncbi.nlm.nih.gov/tools/primer-blast). Primers and accession numbers of the genes used in the study are listed in Table 1. All primers were purchased from Thermo Fisher Scientific.

### 2.11. Statistical Analysis

The results are presented as mean values ± standard deviation (SD). The differences between groups were compared using Student’s t-test. A p value less than 0.05 was considered statistically significant. The statistical analysis was performed, and graphs were prepared using GraphPad Prism 8 software (GraphPad Software, San Diego, CA, USA).

## 3. Results and Discussion

### 3.1. Characterization of Streptococcus thermophilus BGKMJ1-36 and Lactobacillus bulgaricus BGVLJ1-21 Strains

The interest of the dairy industry in production of fermented dairy products is the selection and use of LAB that converts lactose to lactic acid, degrades casein, and produces EPS and antimicrobial compounds, such as organic acids, hydrogen peroxide, antifungal peptides, and bacteriocins [55]. Considering this, we analyzed the physiological, biochemical, and technological characteristics of BGKMJ1-36 and BGVLJ1-21 strains. The results are summarized in Table 2. Both BGKMJ1-36 and BGVLJ1-21 strains have the ability to form curd after 5 h at 42 °C, hydrolyze α_s1_**-**, β**-**, and κ**-** casein, and show antimicrobial activity towards the pathogenic strain *Listeria monocytogenes* ATCC19111. Proteolytic activity of yogurt cultures leads to the production of free amino acids, which may be converted to various flavor compounds, such as ammonia, amines, aldehydes, phenols, indole, and alcohols, all of them contributing to yogurt flavor [3]. LAB can produce various volatile compounds which contribute to the flavor formation of certain dairy products [56]. One of them is acetoin, produced by the BGKMJ1-36 strain. Acetoin is produced by LAB and various microorganisms which degrade glucose and other fermentable carbon sources via the Embden–Meyerhof pathway [57].

These results are in accordance with the fact that *S. thermophilus* and *L. bulgaricus* are LAB widely used as starters in production of fermented dairy products, particularly in yogurts [58]. In addition to these technological properties, we showed that the BGKMJ1-36 strain is an excellent EPS producer (Table 2). The production of EPSs by starter cultures significantly contributes to rheological properties of dairy products, so the characterization of such LAB is of great interest for the dairy industry. Many *S. thermophilus* strains synthesize EPSs that contribute to the desirable viscous texture of fermented dairy products [59]. Production of EPSs could be modified by many factors, e.g., growth medium, temperature, pH, fermentation time, and some other factors [60]. Interestingly, it was shown that *L. bulgaricus* was necessary to ensure better growth of some *S. thermophilus* strains in yogurt, as well as to induce their EPS production [61].

### 3.2. Safety Assessment of Streptococcus thermophilus BGKMJ1-36 and Lactobacillus bulgaricus BGVLJ1-21 Strains

After we showed that the mixed starter culture composed of BGKMJ1-36 BGVLJ1-21 strains have good technological properties, the hemolytic and gelatinase activities of the strains were investigated. No hemolytic activity was observed after 48 h of incubation for BGKMJ1-36, nor BGVLJ1-21. Additionally, both strains showed no gelatinase activity after 48 h of incubation on an agar plate filled with saturated ammonium sulphate solution. The absence of these properties is a favorable characteristic of new starters [62]. Although most dairy bacteria have a long history of use without significant established risk, the high prevalence of antibiotic resistance detected in commercial starter cultures strains raised the demand for novel starters susceptible to clinically relevant antibiotics [63,64]. Taking this into account, in this study, the susceptibility of these strains was tested on recommended concentrations of relevant antibiotics (EFSA, 2018). The results show that strains BGKMJ1-36 and BGVLJ1**-**21 are susceptible to the recommended minimal inhibitory concentrations (MIC) of all tested antibiotics (Table 3). The obtained results prove the safety status of these strains as starters for dairy production.

### 3.3. Acidifying Kinetics of Streptococcus thermophilus BGKMJ1-36 and Lactobacillus bulgaricus BGVLJ1-21 Strains During Milk Fermentation and Yogurt Storage

In addition to characterization of basic technological properties of BGKMJ1-36 and BGVLJ1-21 (Table 2), we further characterized their feature in co-culture in milk, during the laboratory-scale yogurt production. The rate of milk acidification by *S. thermophilus* and *L. bulgaricus* is a technological feature of major interest in yogurt production. For the yogurt production, pasteurized milk was inoculated with 3% starter culture in BGKMJ1-36:BGVLJ1-21 = 1:2 ratio. Since both BGKMJ1-36 and BGVLJ1-21 strains were able to ferment milk individually, we also characterized the pH curve of each strain inoculated individually in milk. BGKMJ1-36 single culture decreased pH value faster (pH 4.78 for 4 h and 45 min) than BGVLJ1-21 single culture (pH 4.82 after 5 h at 42 °C), as well as in comparison to mixed starter culture (pH 4.74 after 5 h at 42 °C) (Figure 1A). On the other hand, the BGKMJ1-36 and mixed culture did not change pH value further, but BGVLJ1-21 single culture decreased the pH value to 4.64 during 24 h. During cooling, the pH value of yogurt was slowly lowered to 4.61 after 7 days of storage at 4 °C, while after 14 days of storage at 4°C, pH value was 4.17, after 21 days, 4.13, and 4.01 after 28 days (Figure 1B). According to Gueimonde et al., post-acidification during storage depends on the LAB used for yogurt production, and greater increases of pH values were found in yogurts that included *L. bulgaricus* [65]. Our results are in accordance with these reports, confirming post-acidification and/or activity of residual microorganisms. In some cases, yogurts fermented with *L. bulgaricus* have been rated by consumers as being too acidic [66]. Hence, further tests at the industrial scale should include different ratios of BGKMJ1-36 and BGVLJ1-21 in order to decrease number of BGVLJ1-21 to avoid the post-acidification.

### 3.4. Growth Kinetics of Streptococcus thermophilus BGKMJ1-36 and Lactobacillus bulgaricus BGVLJ1-21 Strains During Milk Fermentation and Storage

The growth kinetics of BGKMJ1-36 and BGVLJ1-21, in single and mixed cultures, during milk fermentation was evaluated. Species-specific bacterial counts were monitored over 28 days. The CFU of BGKMJ1-36 and BGVLJ1-21 (in single and mixed cultures) exponentially increased during the initial hours of fermentation, and reached a maximum around the fifth hour (Figure 1C: circles, squares, and triangles colored in red). Comparing the counts of bacteria when cultured individually and in combination, BGVLJ1-21 viable cell counts were higher in combination with BGKMJ1-36 (8.3 × 10^8^ CFU/mL), in comparison to CFU when was individually inoculated (3 × 10^8^ CFU/mL). Moreover, the number of viable cells of BGVLJ1-21 in combination with BGKMJ1-36 increased during 24 h cooling time, suggesting that BGKMJ1-36 stimulates growth of BGVLJ1-21 (Figure 1C). In accordance with this, Ranadheer et al. noticed higher counts of viable *L. bulgaricus,* compared to *S. thermophilus,* in yogurt after co-incubation [67]. However, the viability of the BGKMJ1-36 strain decreased during 24 h cooling time. This observation could be explained by hydrogen peroxide production by *L. bulgaricus*, which could partially damage the *S. thermophilus* cells [68]. Also, the symbiotic relationship between *S. thermophilus* and *L. bulgaricus* was described*,* reflected in the fact that *L. bulgaricus* stimulates the growth of *S. thermophilus* when it is in the logarithmic phase of growth. After that, the large amount of lactic acid produced by *L. bulgaricus* inhibits the growth of *S. thermophilus* [6].

Noticeably, the viability of the BGVLJ1-21 started to decrease during the first week of storage. After this time, viable counts of BGVLJ1-21 (1.5 × 10^7^ CFU/mL) remained constant by the end of the 28 days (Figure 1D). The number of viable BGKMJ1-36 cells decreased significantly (37 %) during the second week, and remained constant during 28 days of storage (Figure 1D). These results represent the total counts of cultivable and undamaged cells grown on agar plates of GM17 for BGKMJ1-36 and MRS for BGVLJ1-21 after 24 h at 37 °C. Other authors reported similar results as ours, suggesting that the viability of bacteria depends on strain type, storage conditions, and culture mixture [48,49,69,70].

### 3.5. Effects of Simulated Gastrointestinal Tract Conditions on the Viability of Streptococcus thermophilus BGKMJ1-36 and Lactobacillus bulgaricus BGVLJ1-21 Strains in Yogurt

One of the important features during the examination of starter cultures is the survival of unfavorable conditions in the GIT. The ability of BGKMJ1-36 and BGVLJ1-21 cells to survive in simulated GIT conditions was analyzed (Figure 2A). One-day-old yogurt contained 4.67 × 10^8^ CFU/mL and 8.67 × 10^8^ CFU/mL BGKMJ1-36 and BGVLJ1-21, respectively. After incubation for 90 min in gastric juice, the BGKMJ1-36 CFU decreased to 4.00 × 10^7^ CFU/mL (8.6%), while CFU for BGVLJ1-21 was 5.00 × 10^7^ CFU/mL (5.8%). Results obtained by Soni et al. showed elimination of more than 73% of *L. bulgaricus* NCDC-253 and *S. thermophilus* NCDC-199 during an incubation period of 2 h at pH 2.0 and 3.0. Comparing these data to our results, BGKMJ1-36 and BGVLJ1-21 had higher survival at low pH [71]. The additional decrease in the bacterial counts of both strains was observed after the 10-min incubation period in duodenal juice containing a high bile salt concentration (1%), 1.47 × 10^7^ CFU/mL (3.1%) for BGKMJ1-36, and 3.67 × 10^7^ CFU/mL (4.2%) for BGVLJ1-21. Finally, transfer and incubation for 2 h into intestinal juice (0.3% of bile salt and 0.1% pancreatin) led to an additional decrease in the number of BGKMJ1-36 to 9.33 × 10^6^ CFU/mL (2%) and BGVLJ1-21 to 2.05 × 10^6^ CFU/mL (0.2%). Our results on the viability of BGKMJ1-36 and BGVLJ1-21 strains in the unfavorable GIT conditions are comparable with data available in the literature [72,73]. Many researchers have reported that low pH represents a crucial parameter in the viability of starter cultures [74,75]. It has been shown that milk has an impact as a protector of LAB in simulated GIT conditions and provides better survival [76]. Elizaquível et al. found that starter cultures in yogurt better survive the exposure to low pH and bile conditions, as well as digestive enzymes, than in media without carriers [77].

### 3.6. Adhesion of Streptococcus thermophilus BGKMJ1-36 and Lactobacillus bulgaricus BGVLJ1-21 Strains to Caco-2 Cells After Simulated Gastrointestinal Tract Conditions

Adhesion of LAB to epithelial cells mediates colonization of the GIT, and might be a requirement for the exclusion of pathogen and other beneficial effects on the host [78]. To analyze the adhesion ability of BGKMJ1-36 and BGVLJ1-21, the intestinal phase of yogurt digestion was added to Caco-2 cells. After 2 h incubation of intestinal phase of yogurt digestion on Caco-2 cells, we found that about 5% of BGKMJ1-36 and 35% of BGVLJ1-21 adheres to Caco-2 cells (Figure 2B). Interestingly, while Darilmaz et al. showed that the adhesion index for strains that produce EPSs was higher than in strains that do not produce EPSs [79], our previous results revealed that EPSs produced by *Lactobacillus paraplantarum* BGCG11 hindered its adhesion to intestinal epithelial cells [32], similarly to the results obtained in this study. Namely, the EPS-producing strain BGKMJ-36 adhered to Caco-2 cells with a lower affinity as the EPS-non-producing BGVLJ-21. Overall, both strains showed good binding abilities to Caco-2 cells, which could enable manifestation of their probiotic function. Results obtained by Fernández de Palencia et al. showed that *L. bulgaricus* LBY-27 and *S*. *thermophilus* STY-31 strains had relatively high levels of adhesion to Caco-2 cells of 9% and 5%, respectively. Comparing these data to our results, BGKMJ1-36 had a similar adhesion ability, however, BGVLJ1-21 had higher adhesion ability to Caco-2 cells, and it was 35% [80].

### 3.7. Gene Expression Analysis Revealed the Upregulation of Autophagy and Epithelial Barrier Defense Markers

Considering good adhesion properties of BGKMJ1-36 and BGVLJ1-21 to Caco-2 cells, we further analyzed the probiotic potential of these strains. Autophagy is a key process responsible for maintenance of intestinal physiology by controlling function of the various intestinal cells. Autophagy is important for preventing the invasion and dissemination of pathogens, maintaining barrier integrity, preserving intestinal homeostasis, and controlling mucosal inflammation [81]. Therefore, to investigate whether autophagy could be activated in Caco-2 cells upon BGKMJ1-36/BGVLJ1-21 yogurt treatment, we performed qPCR analysis of genes involved in different steps of the autophagy process, including autophagy induction (*ULK1*), autophagosome formation (*AMBRA, BECN1*, *PIK3C3*, *UVRAG*), autophagosome expansion (*ATG5*, *GABARAP, MAP1LC3B*), and retrieval of autophagic proteins (*SQSTM1*) [82,83]. For most of the autophagy-related genes, we detected significantly higher mRNA levels in yogurt-treated cells in comparison to untreated control (Figure 3A). This result emphasizes that bacteria present in yogurt exhibit the potential to activate autophagy in epithelial cells. These findings are comparable with results from our previous studies reporting that certain *Lactobacillus* strains exhibit the potential to trigger autophagy in different types of mammalian cells [22,23]. However, the application of strains and evaluation of their effects after the transit through simulated GIT have never been tested before. To exclude possibility that activation of autophagy is not a consequence of toxic effect of yogurt, which triggered autophagy, as cell death mechanism [84], we measured LDH release in the media from yogurt-treated cells. We did not notice differences in LDH levels between yogurt-treated cells and control (Figure 3B). This result implies that yogurt did not produce toxicity effect on Caco-2 cells, which is in accordance with strict probiotic regulations demanding that bacterial strains with QPS status should not be toxic or exhibit some deleterious effect on GIT epithelium of the host. 

As autophagy has been linked with the improvement of tight junction barrier function in Caco-2 cells [85], we further analyzed epithelial barrier integrity by checking the expression of the most indicative tight junction markers. Tight junctions (TJs) form the border between the apical and basolateral cell surface responsible for maintenance of intestinal permeability [86]. The most prominent members among these proteins are claudins, ocludin, zonula occludens, and cadherins [87], which are encoded by the *CLDN4*, *ZO-1*, *OCLN,* and *CDH1* genes, respectively. According to the gene expression analysis, we found that yogurt treatment significantly upregulated the expression of all abovementioned genes, suggesting that LAB present in yogurt exhibit the potential to strengthen these adhesion protein complexes (Figure 3C). TJs are often targeted by bacteria, leading to their disruption by pathogens, or increased synthesis in presence of probiotics and their metabolites [21,88]. Moreover, the upregulation of TJs correlates with the induction of autophagy, which is in accordance with the previous results pointing to a strong role of autophagy in controlling paracellular TJ permeability by targeting pore-forming tight junction protein claudin-2 [85].

Moreover, cytokine mediated changes in TJs expression and paracellular permeability, contributing to a different gut-related disorder. It has been showed that pro-inflammatory cytokines (TNF-α, IL-1β, IL-6, IL-8) increase intestinal epithelial tight junction permeability [89], while anti-inflammatory cytokines (TGF-β) [90] could restore function of disrupted barriers. Hence, we followed the expression levels of abovementioned cytokines, but we did not detect expression of pro-inflammatory cytokines (Ct > 35) either in control or yogurt-treated cells, suggesting that yogurt did not cause inflammatory response. In the case of TGF-β, there was no difference in mRNA levels (data not shown), suggesting that upregulation of anti-inflammatory cytokine is not a mechanism which consequently leads to stimulation of TJs expression.

Finally, the important probiotic feature of LAB is the potential to stimulate production of anti-microbial peptides (e.g., defensins), which play important roles in host defense [91]. We followed the expression levels of gene encoding human β-defensin distributed in the mucosal epithelium and skin, thus coming into direct contact with the external environment. Importantly, the results showed increased *DEFB1* mRNA levels after yogurt treatment, compared to control (Figure 3D). Similar results were reported for *Lactobacillus helveticus* SBT2171 (LH2171) strain, which induced the expression of human β-defensin in Caco-2 cells by activating c-Jun N-terminal kinase (JNK) signaling via Toll-like receptor TLR2 [92].

## 4. Conclusions

Altogether, our findings pointed to the fact that yogurt prepared from novel autochthonous strains *S. thermophilus* BGKMJ1-36 and *L. bulgaricus* BGVLJ1-21 has a beneficial effect on the gut barrier by cross-linking the most important processes necessary for maintenance of epithelial homeostasis. To the best of our knowledge, this is the first study showing autophagy-inducing potential of LAB strains used for yogurt preparation. Also, in addition to lactic acid production by these strains, the induction of antimicrobial peptide production by host epithelial cells could be beneficial effects of this yogurt in the case of intestinal infection.

Taken together, these results support the further investigation of the beneficial effects of this mixed starter culture, in order to recommend it for use in the dairy industry as functional starter culture.

## Figures and Tables

**Figure 1 microorganisms-08-01586-f001:**
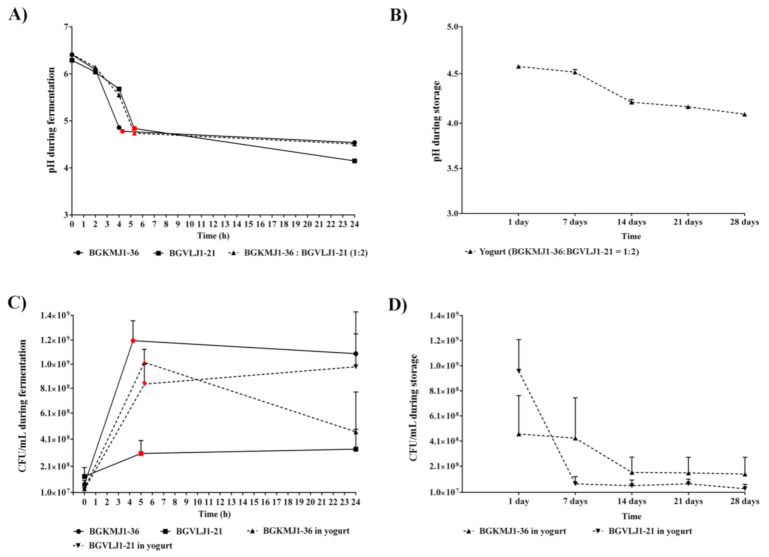
Acidification and growth dynamics of *Streptococcus thermophilus* BGKMJ1-36 and *Lactobacillus bulgaricus* BGVLJ1-21 strains during milk fermentation and storage. (**A**) Acidification curve of BGKMJ1-36 (circle), BGVLJ1-21 (square), and mix of BGKMJ1-36, and in ratio 1:2 (dashed line, triangle) during milk fermentation; (**B**) acidification curve of yogurt during storage at 4 °C; (**C**) growth of BGKMJ1-36 (circle), BGVLJ1-21 (square), individually, as well as in mix in ratio of 1:2 ratio (dashed line, triangle) colony forming unit per milliliter (CFU/mL) during fermentation; (**D**) CFU/mL during storage at 4 °C. Note: circles, squares, and triangles colored in red represent the time of coagulation during milk fermentation.

**Figure 2 microorganisms-08-01586-f002:**
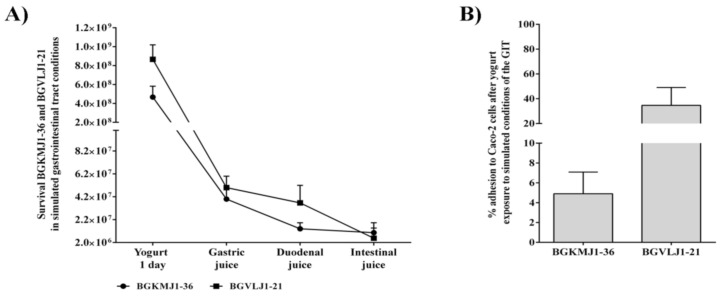
Survival of *Streptococcus thermophilus* BGKMJ1-36 and *Lactobacillus bulgaricus* BGVLJ1-21 in simulated gastrointestinal tract and adhesion to Caco-2 cells after exposure to last phase (intestinal juice) of simulated gastrointestinal tract conditions. (**A**) Survival of BGKMJ1-36 and BGVLJ1-21 in yogurt one-day-old after exposure to simulated gastrointestinal tract (GIT). Gastric juice (pH = 2.0) contained pepsin 0.3%, duodenal juice (pH = 8.0) contained 1% bile salts, while intestinal juice (pH = 8.0) contained 0.3 % bile salts and pancreatin 0.1%; (**B**) adhesion of BGKMJ1-36 and BGVLJ1-21 from yogurt one-day-old after exposure to simulated gastrointestinal tract to Caco-2 cells.

**Figure 3 microorganisms-08-01586-f003:**
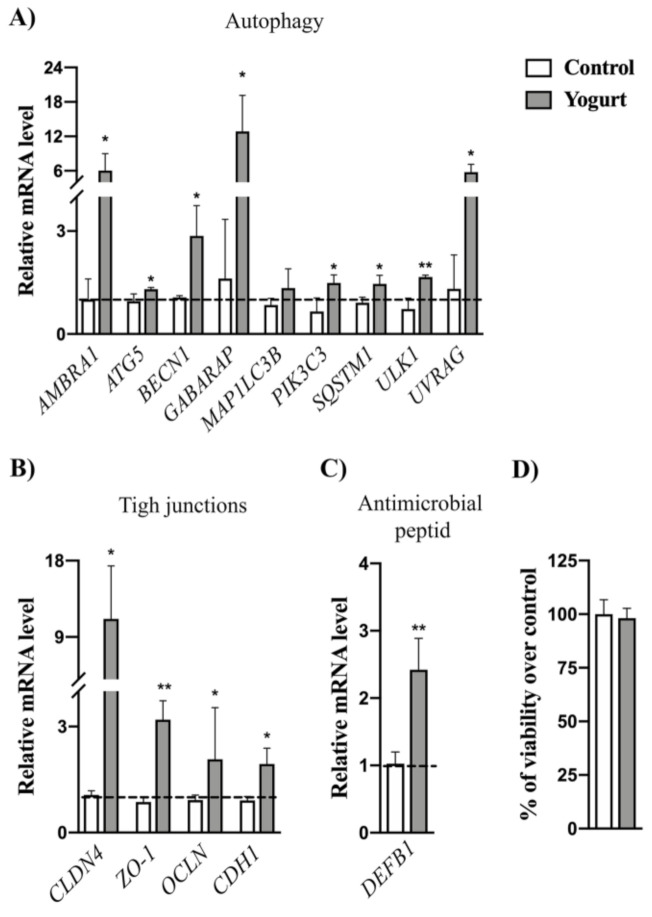
Yogurt application strengthens gut epithelial barrier. Expression of autophagy-related genes (**A**), tight-junction proteins genes (**B**), and human β defensin (**C**) in Caco-2 cells treated with yogurt. Effect of yogurt on Caco-2 cells viability measured by lactate dehydrogenase (LDH) assay (**D**). Student’s t-test was used to compare differences between groups (* *p* < 0.05, ** *p* < 0.01). Caco-2 cells treated with phosphate-buffered saline (PBS) were used as negative control.

**Table 1 microorganisms-08-01586-t001:** List of Primers Used in This Study.

Gene Name	Accession Number of Gene	Primer Sequence 5′–3′	Reference
AMBRA1_F		GGTGGGAGGAGAGGGGATAG	This work
AMBRA1_R	NM_017749.3	CGAGGGGCATGTCATCATTT	
ULK1_F		TTTTGTTTCTCCGTTGGGGC	This work
ULK1_R	NM_003565.4	ACTCTTCCCGGGCTGCTAAT	
UVRAG_F		AGGAAGGAGTGCACTGCAAA	This work
UVRAG_R	NM_001386671.1	AGGCAACTTGACACCGCATA	
GABARAP_F		CCCTCGTCCCGCTGATTTTA	This work
GABARAP_R	NM_007278.2	ATCCCTCCAGCTTGTACCCA	
PIK3C3_F		GCTGTCCTGGAAGACCCAAT	This work
PIK3C3_R	NM_002647.4	TTCTCACTGGCAAGGCCAAA	
MAP1LC3B_F		TTCAGGTTCACAAAACCCGC	This work
MAP1LC3B_R	NM_022818.5	TCTCACACAGCCCGTTTACC	
BECN1_F		CTGGGACAACAAGTTTGACCAT	[44]
BECN1_R		GCTCCTCAGAGTTAAACTGGGTT	
ATG5_F		CACAAGCAACTCTGGATGGGATTG	[45]
ATG5_R		GCAGCCAC GGACGAAACAG	
SQSTM1_F		GCCAGAGGAACAGATGGAGT	[46]
SQSTM1_R		TCCGATTCTG GCATCTGTAG	
CLDN4_F		ACAGACAAGCCTTACTCC	[21]
CLDN4_R		GGAAGAACAAAGCAGAG	
ZO-1_F		AGGGGCAGTGGTGGTTTTCTGTTCTTTC	[47]
ZO-1_R		GCAGAGGTCAAAGTTCAAGGCTCAAGAGG	
OCLN_F		TCAGGGAATATCCACCTATCACTTCAG	[21]
OCLN_R		CATCAGCAGCAGCCATGTACTCTTCAC	
CDH1_F		AGCCTGTCGAAGCAGGATTG	This work
CDH1_R	NM_004360.5	AGAAACAGCAAGAGCAGCAGA	
DEFB1_F		TGTCTGAGATGGCCTCAGGT	[48]
DEFB1_R		GGGCAGGCAGAATAGAGACA	
GAPDH_F		GTGAAGGTCGGAGTCAACG	[49]
GAPDH_R		TGAGGTCAATGAAGGGGTC	
IL-1β_F		TACGAATCTCCGACCACCACTACG	[50]
IL-1β_R		GTACAGGTGCATCGTGCACATAAGC	
IL-6_F		CACTCACCTCTTCAGAACGA	[51]
IL-6_R		CTGTTCTGGAGGTACTCTAGG	
TNF-α_F		AGCCCATGTTGTAGCAAACC	[52]
TNF-α_R		TGAGGTACAGGCCCTCTGAT	
IL-8 F		ACACAGAGCTGCAGAAATCAGG	[53]
IL-8 R		GGCACAAACTTTCAGAGACAG	
TGF-β F		CCGGGTTATGCTGGTTGTACAG	[54]
TGF-β R		AAGGACCTCGGCTGGAAGTGG	

**Table 2 microorganisms-08-01586-t002:** Phenotypic, Biochemical, and Technological Characteristics of *Streptococcus thermophilus* BGKMJ1-36 and *Lactobacillus bulgaricus* BGVLJ1-21.

Feature Tested		BGKMJ1-36	BGVLJ1-21
Shape		Cocci in short and long chains	Longer rods, single and short chains
Catalase test		–	–
Gram staining		+	+
Growth at 15 °C in GM17 broth		–	–
Growth at 37 °C in GM17 broth		+	+
Growth at 45 °C in GM17 broth		+	+
Growth in GM17 broth with 2% NaCl		–	–
Hydrolysis of arginine		–	–
Citrate utilization		–	–
Black zone on bile esculin agar		–	–
Production of CO_2_ from glucose		–	–
Acetoin production (VP)		+	–
Diacetyl production		–	–
Curd forming after		4 h and 45 min	5 h
Exopolysaccharides production		+	–
Aggregation ability		–	–
Hydrolysis of	α_s1_ casein	±	+
	β casein	+	+
	κ casein	±	+
Antimicrobial activity		+ ^a^	+ ^a^

+ positive reaction; – negative reaction; ± weak reaction. ^a^ Both BGKMJ1-36 and BGVLJ1-21 strains inhibit the growth of *Listeria monocytogenes* ATCC19111, but antimicrobial compound is not of proteinaceous nature (Appendix A). Note: BGKMJ1-36 and BGVLJ1-21 strains are sensitive towards antibiotic suggested according to EFSA (2018).

**Table 3 microorganisms-08-01586-t003:** Minimal Inhibitory Concentrations (MIC) of Nine Antibiotics on *Streptococcus thermophilus* BGKMJ1-36 and *Lactobacillus bulgaricus* BGVLJ1-21.

Antibiotic	BGKMJ1-36	BGVLJ1-21
Ampicillin	≤ 1 (2)	≤ 1 (2)
Vancomycin	≤ 0.5 (4)	≤ 0.5 (2)
Gentamicin	≤ 16 (32)	≤ 8 (16)
Kanamycin	n.r.	≤ 4 (16)
Streptomycin	≤ 32 (64)	≤ 8 (16)
Erythromycin	≤ 1 (2)	≤ 0.5 (1)
Clindamycin	≤ 0.5 (2)	≤ 0.5 (4)
Tetracycline	≤ 2 (4)	≤ 0.5 (4)
Chloramphenicol	≤ 1 (4)	≤ 1 (4)

Breakpoints (mg/L) suggested by EFSA (2018) for *Streptococcus thermophilus* and *Lactobacillus bulgaricus* strains are reported in parentheses; note: n.r.: not required.

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
