# Peer review of "Yogurt Produced by Novel Natural Starter Cultures Improves Gut Epithelial Barrier In Vitro"

_microorganisms, 2020, doi:10.3390/microorganisms8101586_

Round 1

Reviewer 1 Report

Comments and suggestions for authors

The authors of the manuscript were interested in two novel candidate probiotic strains (S. thermophilus BGKMJ1-36 and L. bulgaricus BGVLJ1-21) for yogurt production (technological point-of-view) and the potential health-promoting effects (enhancement of the intestinal epithelial barrier, and antimicrobial activity against L. monocytogenes) associated to its consumption, initially performed in vitro. According to this study, BGKMJ1-36 and BGVLJ1-21 were classified as being potentially safe in terms of antimicrobial resistance, displayed good adhesion to Caco-2 cells, and triggered the expression of autophagy-related genes after have successfully survived the unfavorable gastrointestinal tract conditions.

Overall, the introduction section is short and could be strengthened from more information about the current use of indigenous microorganisms as starter cultures. Around 60% of the references used in this section have more than 5 years of their publications, not covering the current state-of-the-art. The technological characterization of BGKMJ1-36 and BGVLJ1-21 for yogurt manufacturing is well represented and discussed, however, there is no mention of the genomic features of the microorganisms involved in this study, an essential analysis for bacterial strains proposed for probiotic use. Lastly, the use of positive control for autophagy assay could better support the main findings.

Specific comments

Abstract

  • Line 16. Please, clarify if throughout the manuscript the strains will be named by their letters and numbers.
  • Lines 18-19. EPS production was evaluated qualitatively (detected visually). Is this result enough to claim that the exopolysaccharide produced by the strain BGKMJ1-36 contributes to the rheological properties of the yogurt?

Introduction

  • Lines 47 – 50. Whole genome sequence (WGS) analysis should be added as one of the requirements for the evaluation of an effective probiotic. I suggest the following manuscripts as reference: https://doi.org/10.3389/fmicb.2019.00424

https://doi.org/10.2903/j.efsa.2018.5206

Materials and methods

  • Lines 82 – 83. Please, provide strains accession numbers.
  • Line 109. Please mention the positive controls used in these assays?
  • Line 116. What was the range of concentration of the antibiotics used in this test? What was the final CFU per well? Was there any positive control? Why a Plate Reader Infinite 200 pro was used to determine the MIC value?
  • Line 123. Why only Gram-positive indicator strains were considered in this analysis?
  • Line 138. Please add its composition.
  • Lines 156. Please describe the composition of the gastrointestinal juices used in this assay.
  • Line 162. Which is the initial count (CFU/mL)?
  • Line 166. Fetal Bovine Serum
  • Line 170. Which is the total number of viable cells in this volume? It’s not clear whether cells suffered or not any washing before Caco-2 cells treatment.
  • Line 190. Please cite the manuscript describing the 2-ΔΔCt
  • Line 192. There is no information concerning the strategy adopted for primer design in this study. Please describe it.
  • Lines 204 and 217. Please clarify the meaning of “basic physiological” and “basic technological”.
  • Line 208. Awkward. Please consider rephrase.
  • Lines 217-218. Is this assumption only based on a table reporting a positive score? BGKMJ1-36 is an excellent EPC producer in comparison with each strain? Was there any EPS quantification?
  • Lines 229. Could be interesting to add a supplementary figure showing the inhibitory zone result of the non-proteinaceous compound activity.
  • Line 230. A table with the MIC values obtained for each antibiotic and the cut-offs established by EFSA could better represent this result.
  • Line 235. I suggest using the word “hemolytic” throughout the manuscript.
  • Lines 243-244. The authors do not report any MIC value.
  • Lines 299-300. Please add the magnitude of reduction in % or log.
  • Lines 318-319. There is a lack of comparison with other studies and a conclusion statement. Are these good or not strains in terms of survivability to the unfavorable GIT conditions when compared with other candidates? I suggest the following manuscript as reference: https://doi.org/10.1016/j.jff.2019.02.004
  • Lines 339-340. Again, there is a lack of comparison with additional studies. To claim that the strains used in this study have good binding abilities to Caco-2 cells, further comparisons must be made.
  • Line 348. Why the authors did not evaluate pro- and anti-inflammatory cytokines on the cell culture supernatant or by RT-qPCR?
  • Line 355. What was the negative control used in this analysis?
  • Lines357-358. Why not use one of these strains or a chemical compound as a positive control?

References

  • Many scientific names are not italicized.

Author Response

All corrections in the manuscript, according to reviewers suggestions, are provided by Track Changes option. Suggestions for new references, as well as replacement of references are accepted. Zotero (a tool that collects, organizes and cite references) was used to edit the references. Additionally,  we added a supplementary figure that showing the inhibitory zone result of the non-proteinaceous compound activity.

Reviewer 2 Report

  • General: please, review the Materials and Methods slightly to make them more reproducible for other authors
  • Specific:
  • Paragraph 2.6: Please describe better the process for yoghurt production, and the storage conditions (e.g. yogurt box); the sampling was made from the same batch? therefore, O2-uptake was possible during the storage, or the yogurt was previously aliquoted?
  • Paragraph 2.7. Why did the authors test the survival to simulated GIT only on 1-day yogurt? The survival after 28 ° C of storage is equally important, as the GIT tolerance may be modified by cold-starvation conditions.
  • Same consideration for the paragraph 2.8.
  • Please specify the substrates for colony counts and survival
  • Please note that the proportion of survived cells (total counts) is referred only to “cultivable and undamaged cells”, otherwise the authors should have used substrates for the recovery of VBNC and damage cells. Please specify in Results sections (discussion for Figure 1). The long-term storage, in fact, may induce the VBNC and/or damage status.
  • Figure 2: I confirm my perplexity on the assay performed only on 1-day yoghurt and not on the other samples stored up to 28 days at 4 ° C.
  • Table 1: please add the relative genes near the primer sequences
  • Figure 3A: the scale of the y-axis is not clear for me ....

Author Response

Thank you very much for your comments. We have corrected everything you have suggested.

Our answers are in the text below.

Round 2

Reviewer 1 Report

Minor concerns

Line 139 - 5x106

Line 159 - (1 M NaH2PO4 and 1 M Na2HPO4)

Lines 202-203 - Please add the multiplicity of infection (MOI) used on this assay. Is there any reason why an MOI around 2 was used? I also suggest the inclusion of the information saying that recovered cells were washed with PBS buffer before Caco-2 cells treatment. 

Line 224 - Please provide the accession numbers of the genes used in this study for primer design.

Author Response

Thank you very much for your comments. We have corrected everything you have suggested.
